# The Emerging Role of MXenes in Cancer Treatment

**DOI:** 10.3390/ijms262110296

**Published:** 2025-10-22

**Authors:** Najla M. Salkho, William G. Pitt, Ghaleb A. Husseini

**Affiliations:** 1Department of Chemical and Biological Engineering, College of Engineering, American University of Sharjah, Sharjah P.O. Box 26666, United Arab Emirates; nsalkho@aus.edu; 2Materials Science and Engineering Program, College of Arts and Sciences, American University of Sharjah, Sharjah P.O. Box 26666, United Arab Emirates; 3Chemical and Biological Engineering Department, Brigham Young University, Provo, UT 84602, USA; pitt@byu.edu

**Keywords:** MXenes, 2D nanomaterials, photothermal therapy, cancer treatment, biocompatibility

## Abstract

MXenes are relatively new 2D materials made up of carbides and/or nitrides of transition metals with a chemical formula M_n+1_X_n_T_x_. They are usually fabricated by chemically etching a ceramic phase. MXenes possess tunable catalytic, optical, and electronic properties, which have attracted significant research interest, primarily in energy storage and biosensing applications. Since their first fabrication in 2011, there has been a rapid increase in studies investigating the use of MXenes in a wide range of applications. In this review, the synthesis methods of MXenes are discussed. Then, the potential application of MXenes in cancer treatment is highlighted based on current research. The ability of MXene to convert light, usually NIR (I and II), to heat with improved conversion efficiencies makes it a competitive candidate for photothermal cancer therapy. Moreover, the surface of MXenes can be modified with drugs or nanoparticles, thereby achieving synergistic photo/chemo/, and sonodynamic therapy. This review also examines the available research on the biocompatibility and cytotoxicity of MXenes.

## 1. Introduction

According to the World Health Organization (WHO), cancer resulted in around 9.7 million deaths worldwide in 2022 [1]. In addition, there were nearly 20 million new cancer cases diagnosed in 2022, and the figures are expected to reach 28 million cases worldwide by 2040, according to Cancer Research, UK [2]. Normal human cells undergo growth and multiplication through the process of cell division. When cells age or become damaged, they die; however, new cells are generated in a controlled manner to replace them. Hundreds of genes regulate the process of cell division. A balance between the genes responsible for cell proliferation and suppression is essential to maintain the normal cycle of cell growth; in addition to suppression, there are genes responsible for the programmed cell death, known as apoptosis [3]. When mutations occur in the genes that control cell growth, these cells may become non-responsive to suppression and apoptotic signals; instead, they proliferate in an uncontrolled manner. Eventually, they may form a tumor that can either be benign or cancerous. The latter is characterized by aggressive invasion into other body tissues, known as metastasis.

Mustard gas was the first chemotherapeutic agent used to treat cancer in 1943 by Louis Goodman and Alfred Gilman after its use in World War I [4]. They collaborated with surgeon Gustaf Lindskog at Yale University to use Mustine (nitrogen mustard) to treat a patient with non-Hodgkin’s lymphoma. The response was positive at that time, marking the onset of the use of cytotoxic agents, now known as chemotherapy. However, despite the synergistic effect of chemotherapy when combined with other treatment modalities, such as surgery or radiotherapy, the side effects of chemotherapy constitute severe roadblocks and are often regarded as traumatic for many cancer patients. Thankfully, the rise of drug delivery systems (DDSs) has significantly minimized the side effects of chemotherapy. Carriers at the nanoscale size are being extensively examined in research to gauge their application in DDSs. These carriers include micelles, liposomes, dendrimers, and solid–lipid nanoparticles, among others.

This review will discuss a relatively young class of 2D materials known as MXenes to assess their use in cancer treatment. MXenes are 2D carbides or nitrides of transition metals with a general formula of M_n+1_X_n_T_x_, where M is an early transition metal (Figure 1); X is carbon and/or nitrogen; n is the number of layers of carbon/nitrogen, which is usually between 1 to 3; and T_x_ is a surface terminator such as F, Cl, O, OH [5]. Titanium carbide (Ti_3_C_2_T_x_) was the first MXene discovered at Drexel University in 2011 [6] and the one that is commonly used in research. The name “MXene” was coined by Naguib and coworkers to emphasize the graphene-like morphology of the newly developed 2D material [6].

MXenes are prepared from a MAX phase, where A is an element from groups 13–16 in the periodic table. MAX phases are strongly bonded layered structures [7] present in the following three structures M_2_AX (n = 1), M_3_AX_2_ (n = 2), and M_4_AX_3_ (n = 3). The A element in the MAX phase is bonded to the M element via a metallic bond, which is usually removed by wet etching with hydrofluoric acid (HF) to prepare MXenes (M_n+1_X_n_T_x_), and the resulting bond M-X could be metallic, covalent, or ionic. MXenes are recently considered hot topics in research compared to other 2D materials owing to their high electric conductivity, strength, stability, hydrophilicity, large surface area, and other tunable properties that enable their use in many applications, such as energy storage, electrocatalysis, and photothermal therapy [7,8].

## 2. Synthesis of MXenes

MXenes are often prepared following the top-down approach, starting with a 2D MAX phase. There are various synthesis techniques; the common ones include: HF/HF in situ etching, molten salt etching, alkali-assisted hydrothermal etching, and chemical vapor deposition (CVD). Unlike the former etching methods, CVD is a bottom-up approach that does not require a MAX phase to initiate MXene preparation.

### 2.1. Hydrofluoric Acid Etching

Etching the MAX phase with HF is the standard approach followed in most studies. Common A elements in the MAX phase include silicon (Si), germanium (Ge), tin (Sn), indium (In), and aluminum (Al) [9]. Among these elements, an Al-containing MAX precursor is more favorable to start with because aluminum has the highest chemical reactivity and the lowest reduction potential, thus making the M-Al bond the easiest to cleave. Naguib and coworkers [6] were the first to synthesize Ti_3_C_2_T_x_ MXene by etching Ti_3_AlC_2_ (MAX phase) with concentrated HF (50%) according to the reactions below:Ti_3_AlC_2_ + 3HF → AlF_3_ + 3/2H_2_ + Ti_3_C_2_(1)Ti_3_C_2_ + 2H_2_O → Ti_3_C_2_(OH)_2_ + H_2_(2)Ti_3_C_2_ + 2HF → Ti_3_C_2_F_2_ + H_2_(3)

The procedure involved mixing Ti_3_AlC_2_ powder with HF (50%) at room temperature for 2 h, after which the suspension was washed several times with deionized water until reaching a neutral pH. HF was found to selectively remove the Al layer in the MAX phase. This resulted in a weakly bonded multi-layer structure that was delaminated by sonication to yield separate layers of atomic thickness. Ti_3_C_2_ MXene fabrication by HF etching resulted in a mixed functionalized surface with -OH and -F terminators, as identified from the XRD pattern and XPS spectra, as reported by Naguib et al. [6]. The spacing between the layers in MXenes significantly contributes to their electronic and physical properties; this spacing can be controlled by intercalating molecules between the layers. In the case of HF etching, water molecules are responsible for the intercalation process. HF etching produces multilayered MXenes, where the layers are bonded through Van der Waals interactions and hydrogen bonding [10]. To expand the spacing between layers, intercalating agents are used, followed by ultrasound to exfoliate the material into single- or few-layer nanosheets [10]. In other synthesis methods, intercalation is achieved by cations (e.g., Na^+^, K^+^, NH_4_^+^) or organic molecules (e.g., tetramethylammonium hydroxide, tetrabutylammonium hydroxide, dimethyl sulfoxide). The structure of the MAX phase and the etching conditions, including HF concentration, temperature, and etching duration, can influence the structure and properties of the resulting MXene. For example, using a lower concentration of HF requires a longer immersion time to etch the A element completely from the MAX phase. At high HF concentration, the prompt liberation of the H_2_ gas (Equation (1)) results in a unique accordion-like structure in SEM images (Figure 2d), unlike the compact structure of the MAX phase (Figure 2c), and hence verifies the synthesis of MXenes [11]. However, the absence of this structure (stacked layers) under SEM when etching at low HF concentrations does not necessarily dispute the synthesis of MXenes, especially if sufficient time is allowed for exfoliating the A layers. For further information, please refer to the study reported by Alhabeb et al. [11].

The etching time of the MAX phase under certain experimental conditions must be optimized to completely extract the A element, but without over-etching. As an example, Wang et al. [12] examined the effect of etching time on the capacitive properties of Ti_3_C_2_T_x_. The synthesis was performed at room temperature using Ti_3_AlC_2_ as the MAX phase etched with 40% HF. The authors concluded that a complete etching of the Al layers required 4 days, as verified by ADF-STEM images and XRD patterns. In XRD patterns, the (002) peak of Ti_3_C_2_T_x_ shifted to lower angles as etching time increased up to 4 days and remained at a constant angle afterwards up to 17 days. This indicates an increase in the lattice parameter *c* due to etching the Al atoms. Moreover, the (104) peak of Ti_3_AlC_2_ at 39.2° was diminished by 24 h of HF etching, thus supporting the previous findings. Over-etching of Ti_3_C_2_T_x_ was also observed by Wang and coworkers [12] after 17 days, where Ti started to dissolve, leading to the formation of micro-cracks perpendicular to Ti_3_C_2_T_x_ layers.

**Figure 2 ijms-26-10296-f002:**
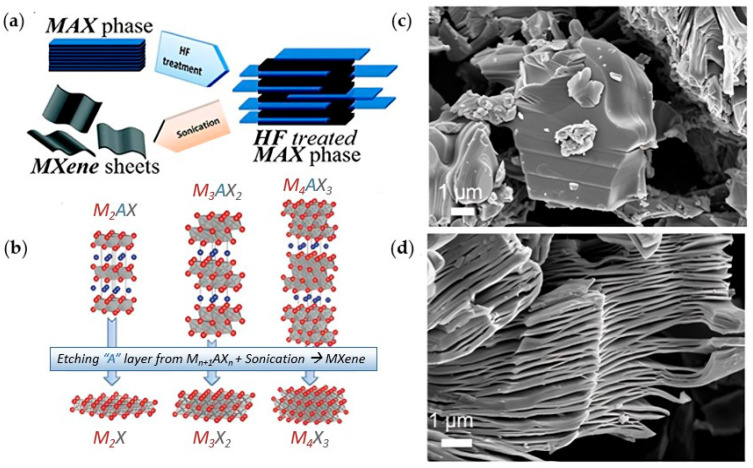
(**a**) Exfoliation of MAX phase using HF to form (**b**) MXenes multi-layers. (**a**,**b**) are reproduced from [13,14] with permission from PubMed Central and the American Chemical Society. SEM images of (**c**) Ti_3_AlC_2_ (MAX phase) and (**d**) Ti_3_C_2_T_x_ (30 wt% HF); reproduced from [11] with permission from the American Chemical Society.

HF etching is considered one of the fastest methods for synthesizing MXenes; however, one cannot underestimate the serious health hazards associated with HF acid. HF acid is highly corrosive and can penetrate skin deeply, causing irreversible burns and tissue necrosis even at low dosages [15]. One way to overcome the direct use of HF acid is to form it in situ by reacting hydrochloric acid (HCl) with a fluoride-containing salt (e.g., LiF, NaF, KF, NH_4_F) as first reported by Ghidiu et al. [16].

### 2.2. Molten Salt Etching

Nitride MXenes are relatively less explored in the literature, but they are expected to have superior properties over their carbide counterparts. A comprehensive study was conducted by Zhang et al. [17] to compare MXenes’ properties using density functional theory calculations. The study reflected enhanced electrical conductivity of nitride-based MXenes compared to carbide-based MXenes [17]. In addition, the unique bond between the transition metal and the nitrogen in Ti_n+1_N_n_ resulted in stronger adhesion to terminal groups, signifying more active sites that are suitable for catalysis applications [17]. While HF etching is the conventional method for synthesizing carbide-based MXenes, it cannot be used to fabricate nitride MXene (Ti_n+1_N_n_), as the MAX phase dissolves in the HF solution during the process. The Al atoms are tightly bonded to the N atoms in the Ti_n+1_AlN_n_ MAX phases, thus resulting in a high energy of formation compared to the C-based MAX phases [9]. In addition, the low cohesive energy of nitride-based MAX phases renders them less stable in HF solutions [9]. To bypass the challenges of HF etching for N-based MAX phases, a molten salt etching process was introduced to fabricate nitride MXenes. Urbankowski and coworkers [18] were the first to explore molten salt etching using a combination of fluoride salts (KF, LiF, NaF) to fabricate Ti_4_N_3_ MXene. The synthesis begins by mixing and ball milling the powdered salts of KF, LiF, and NaF at their eutectic composition (59:29:12 wt%) with Ti4AlN3 (MAX phase) in a 1:1 mass ratio. Mixing salts at the eutectic composition is advantageous, as it enables etching at low temperatures, thereby preventing MXenes from developing structure defects that would otherwise occur at elevated temperatures. The powdered mixture was then treated in an alumina crucible at 550 °C for 30 min in the presence of Ar purging gas. After etching, Al-containing fluorides (Na_3_AlF_6_, K_2_NaAlF_6_, K_3_AlF_6_, AlF_3_, and LiNa_2_AlF_6_) were separated from MXenes by washing with 4 M H_2_SO_4_. A sequence of washing with DI water followed by centrifugation was repeated until the decanted supernatant was near neutral pH. Then, the resulting sediment composed of multilayered Ti_4_N_3_ MXene was filtered and delaminated into fewer layers of flakes by mixing and handshaking in 40 wt% tetrabutylammonium hydroxide (TBAOH). Finally, TBAOH was removed from the supernatant by rinsing and centrifugation with DI water. To separate smaller flakes, Ti_4_N_3_ powder was sonicated in DI water, and flakes were separated from the supernatant by filtration.

Molten salt etching can also be used to fabricate carbide-MXenes in addition to the conventional HF etching method discussed in the previous section. However, the synthesis of Ti_3_C_2_ MXenes using the molten salt method produces hydrophobic MXene clay due to the absence of -OH terminators. This poses a challenge as MXene flakes become non-dispersible in water (a solvent often used in the synthesis protocol). For example, Kamysbayev et al. [19] fabricated Ti_3_C_2_T_x_ nanosheets using CdBr_2_ and CdCl_2_ molten salts. The formed MXene was dispersible in N-methylformamide (NMF), but not in water. Hence, this limits the use of MXenes in applications involving aqueous media. Arole et al. [20] first reported the synthesis of water-dispersible Ti_3_C_2_T_x_ nanosheets using molten salt (SnF_2_). In molten salt etching, multiple salts are typically used at the eutectic composition to lower the melting point, thereby enabling the reaction to proceed below 800 °C, the temperature above which MXene starts to degrade. However, Arole and coworkers [20] were able to find a single salt (SnF_2_) that has a low melting temperature of 213 °C, thus bypassing the use of multiple salts in the conventional protocol. After etching was completed, the clay containing Ti_3_C_2_T_x_ was washed with KOH to remove unreacted SnF_2_ and render MXene dispersible in water due to the addition of -OH terminal groups. However, even with this synthesis protocol, etching must be conducted under Ar purging (which adds extra cost to the synthesis) to preclude the oxidation of MXene.

Chen et al. [21] adapted the molten salt-shielded synthesis (MS^3^) protocol, originally used to fabricate the MAX phase, and combined it with Lewis salt etching to prepare MXenes. The “one-pot” method, as termed by the authors, has enabled the synthesis of Ti_3_C_2_T_x_, Ti_2_CT_x_, Ti_3_CNT_x_, and Ti_4_N_3_T_x_ MXenes in air without the need for inert gas protection. The synthesis starts by pelletizing the MAX phase (e.g., Ti_3_AlC_2_) with NaCl/KCl. The pellet was then immersed in a mixture of NaCl/KCl/CuCl_2_ inside a crucible and heated to 700 °C in a muffle furnace under air for 10–40 min. Upon heating, the eutectic salt bath (NaCl/KCl/CuCl_2_) began to melt at 300 °C, thus providing a shielding blanket that separates the MAX pellet from the air. Etching began at 700 °C, where copper was reduced from Cu^2+^ to Cu and Al was oxidized in the Ti_3_AlC_2_ MAX according to the reaction in Equation (4) [21]. After etching was completed, the crucible was cooled, and the sample was washed with deionized water followed by a solution of ammonium persulfate (0.5 M) to remove the solidified salt and the reduced Cu particles, respectively.Al + 3/2CuCl_2_ → AlCl_3_ + 3/2Cu(4)

### 2.3. Other Synthesis Approaches

The wet chemical etching methods mentioned previously are top-down approaches that are predominantly used in MXenes fabrication. However, wet chemical etching often uses hazardous reagents, requires high energy consumption, and is often considered a trial-and-error method to optimize the synthesis conditions, depending on the parent MAX phase. The synthesis of MXenes using bottom-up approaches, albeit less investigated in the literature, offers meticulous control of the MXene chemistry, hence eliminating structure defects and tailoring its properties for the intended application. Chemical vapor deposition (CVD), plasma-enhanced pulsed laser deposition, and solid-state direct synthesis are bottom-up approaches used in MXenes preparation. We will limit our discussion to the CVD method in this section. In CVD, gaseous precursors, instead of a MAX phase in wet chemical etching, are reacted on a substrate.

#### Synthesis of MXenes by Chemical Vapor Deposition (CVD)

Xu et al. [22] first reported the synthesis of ultrathin *α*-Mo_2_C MXene using a CVD process. The fabricated MXene exhibited a crystal structure with regular shapes (triangles, rectangles, hexagons, octagons, nonagons, and dodecagons) and varied lateral sizes, ranging from 10 to over 100 μm, and thicknesses of 3–20 nm. Briefly, the CVD process started with a substrate (5 × 5 mm^2^) of Cu foil sitting on a Mo foil. The substrate was placed in a quartz tube heated to above 1085 °C inside a horizontal tube furnace under H_2_. Then, a flow of CH_4_ was introduced at ambient pressure to start the growth of the 2D *α*-Mo_2_C MXene for 2–50 min. To prevent the formation of graphene in this process, the methane flow was carefully controlled at a low concentration. In addition, a rapid cooling rate was required after growth to ensure the formation of clean and defect-free ultrathin *α*-Mo_2_C crystals. After rapid cooling of the MXene to room temperature, the samples were transferred in a manner similar to the transfer of graphene [23]. Crystals were found to exhibit excellent thermal and chemical stability in air for a few months. Also, the CVD was operated under ambient pressure instead of the ultrahigh vacuum (UHV) condition required for high-quality 2D superconducting films. Under an optical microscope, *α*-Mo_2_C crystals with thicknesses of 6–11 nm reflected different colors on SiO_2_/Si substrates, thus enabling rapid thickness identification.

Wang et al. [24] grew carpet-like sheets of Ti_2_CCl_2_ by reacting gaseous CH_4_ and TiCl_4_ diluted in Argon at 950 °C on a Ti substrate. They also fabricated MXenes that have not been produced previously by conventional etching methods. For example, Zr_2_CCl_2_ and Zr_2_CBr_2_ MXenes were produced by CVD using CH_4_ and ZrCl_4_ or ZrBr_4_ precursors in the gas phase over a Zr foil at 975 °C. The growth of the Ti_2_CCl_2_ carpet-like structure (Figure 3a,b) via CVD showed unique features as the thickness of the sheets increased. As the reactants need to diffuse through the carpet sheets to reach the substrate, it was expected that the reaction would slow down and eventually cease when the thickness of the MXene sheets increases. However, the carpet-like structure wrinkled into bulges that detached as vesicles due to the in-plane stress, as shown in SEM images in Figure 3c–e, leaving the substrate behind exposed to the precursors. This enabled the continuous production of MXenes with maximum utilization of the substrate.

## 3. Applications of MXenes in Cancer Treatment

MXenes, like other 2D nanostructures, are expected to cause an improved lateral accumulation via vessels in tumors due to more tumbling and rolling dynamics during flow in the bloodstream [8]. In addition, MXenes possess a high surface area with attractive terminators (e.g., -OH, -O, -Cl, -F), thus allowing surface functionalization for possible micro and nanostructures with superior targeting specificity [8]. Surface terminators such as -OH endow MXenes with hydrophilicity and adsorption capability for positively charged molecules that can be employed as, or to deliver, cationic drugs [25,26]. Cancer nanotheranostics is another promising field for MXenes due to their magnetic and electric properties [8]. MXenes can be loaded with chemotherapeutic agents such as doxorubicin (DOX) due to the strong π−π stacking interaction, resulting in a high drug loading capacity and responsiveness to pH and NIR irradiation in drug release [8]. Furthermore, the high light-to-heat conversion of MXenes, attributed to localized surface plasmon resonance (LSPR) phenomena, makes them appealing for photothermal therapy (PTT) [25]. Synergistic effects can also be achieved using MXenes in chemo-photothermal therapy [25,27,28]. Photothermal therapy (PTT) relies on photothermal conversion to kill tumor cells. In brief, when electrons are excited, they release excess energy in the form of heat as they return to their ground state. This heat raises the temperature of the tumor, thus causing biochemical and morphological changes in the tumor cells. The extent of damage depends on the temperature reached. At low to moderate temperatures (41–43 °C), the heat may not directly kill tumor cells, but it can induce protein aggregation and denaturation [29]. Additionally, it can increase vascular permeability near the tumor, facilitating the transport of drugs and oxygen to the affected area. At moderate temperatures (43–45 °C), heat has a significant impact on tumor cells through partial protein denaturation, cytoskeletal reorganization, and regulation of gene expression, generating reactive oxygen species (ROS) [29]. At elevated temperatures (45–55 °C), tumor cells experience substantial thermal stress, leading to the leakage of intracellular contents and triggering programmed cell death through heat-induced apoptosis and necrosis [29].

The first paper reporting the successful fabrication of MXenes was published in 2011 by Naguib et al. [6]. Since then, the number of publications on MXenes has been increasing exponentially since 2015, as shown in Figure 4. Energy storage, sensors, and water treatment are the primary applications of MXenes in research; however, their use in cancer treatment has started to attract attention, as indicated by publication metrics retrieved from the Dimensions database in Figure 4 [30]. The relatively low share of MXenes in biomedical applications compared to others may be attributed to the unexamined aspect of biocompatibility.

Han et al. [28] were the first to investigate the application of Ti_3_C_2_ MXene in cancer treatment by exploiting the chemo-photothermal synergy. Ti_3_C_2_ MXene was prepared by mixing Ti_3_AlC_2_ with 40% HF at room temperature for 3 days. The aggregated nanosheets were separated by intercalation with 25 wt% of tetrapropylammonium hydroxide (TPAOH), which affects the size of the nanosheets according to the treatment duration. Then the Ti_3_C_2_ nanosheets were centrifuged, washed with ethanol, deionized water, and stored in ethanol at 4 °C. Ti_3_C_2_ MXene nanosheets are dispersible in pure water; however, they aggregated in physiological solutions (containing salts). This issue can be resolved by treating the surface of Ti_3_C_2_ with polyethylene glycol (PEG) [31,32], polypropylene glycol (PPG) [31], or soybean phospholipid (SP) [28,33]. Han and coworkers [28] treated the surface of Ti_3_C_2_ nanosheets with soybean phospholipid (Ti_3_C_2_-SP) using a vacuum rotary evaporator to improve their biocompatibility. Then Dox was loaded onto Ti_3_C_2_-SP by sonication and mixing for 24 h at room temperature to produce Dox@Ti_3_C_2_-SP. The electrostatic interaction between the negatively charged Ti_3_C_2_-SP (zeta potential = −41.1 mV) and the positively charged Dox molecules promoted the loading, which was confirmed by the shift in zeta potential from −41.1 to −28.9 mV for Ti_3_C_2_-SP compared to Dox-loaded Ti_3_C_2_-SP. According to dynamic light scattering (DLS), the planar size of the Ti_3_C_2_ sheets was found to be 122.4 nm with a thickness of 0.9 nm. The loading of Dox increased the planar size up to ~200 nm, which is still within the desired range for the enhanced permeability and retention (EPR) effect, a phenomenon independently discovered by Jain and Maeda in 1986 [34,35]. The EPR effect describes the defective nature of tumor blood vessels, characterized by gaps (~2000 nm) [36] between endothelial cells. This leaky vasculature allows the passive transport of nanoparticles from the tumor blood vessels into the interstitial space of tumor tissues, where they are retained due to inadequate lymphatic drainage in solid tumors. The sensitivity of Dox@Ti_3_C_2_-SP to pH was assessed by drug release studies in dialysis bags at 3 different pH levels (4.5, 6, 7) for 12 h. The highest drug release of 57.9% was found at a pH of 4.5, followed by that at a pH of 6 (33.9%) and a pH of 7 (14.2%). This confirms the responsiveness of the loaded MXene to pH that simulates the acidic condition in many tumor microenvironments.

In the same paper by Han and coworkers [28], the synergy between the PTT and the chemotherapeutic effect of Dox@Ti_3_C_2_-SP was also investigated in vitro using murine breast cancer 4T1 cells with the aid of confocal laser scanning microscopy (CLSM). For the photothermal studies, groups were irradiated with NIR at 808 nm and 1.5 W/cm^2^ for 5 min. After incubating the 4T1 cells with various treatments for 4 h, the cells were co-stained with Calcein-AM and propidium iodide (PI). The green fluorescence of Calcein-AM in the CLSM images marks the live cells, while the red fluorescence of the PI indicates the dead cells, as shown in Figure 5a. The photothermal effect (Ti_3_C_2_-SP+Laser) induced significant cell death. Interestingly, the combination of photothermal and chemotherapy treatment resulted in the highest cell death (Dox@Ti_3_C_2_-SP+Laser), as shown in Figure 5b.

Han et al. [28] also reported the results from in vivo studies on 4T1 tumor-bearing mice using 5 treatments administered intravenously: saline (control), NIR-irradiation only, free Dox (3 mg kg^−1^), PTT+Ti_3_C_2_-SP (15 mg kg^−1^), synergistic PTT and Dox@Ti_3_C_2_-SP (15 mg kg^−1^ = 3 mg kg^−1^ Dox). The PTT-treated groups were irradiated by NIR laser at 808 nm, 1.5 W/cm^2^ for 10 min after 4 h from the IV injection. PTT combined with Ti_3_C_2_-SP showed 80.3% tumor inhibition, but with tumor recurrence. However, the PTT and Dox@Ti_3_C_2_-SP synergy resulted in complete tumor eradication with no recurrence [28]. IR images from a thermal imaging camera showed a significant increase in temperature (up to 68.5 °C) in PTT-treated groups (808 nm, 1.5 W/cm^2^ for 10 min), which resulted in tumor ablation.

The light-harvesting capability of Ti_3_C_2_ nanosheets can be further improved. Liu et al. [25] reported a high photothermal conversion efficiency of ~58.3% (808 nm irradiation) when Ti_3_C_2_ nanosheets (~100 nm) were modified with Al oxoanion group Al(OH)_4_^−^ on their surface. The synthesis involves etching the Ti_3_AlC_2_ phase with 9M HCl and LiF in the presence of AlCl_3_.6H_2_O for 3 days, followed by intercalation with tetramethylammonium hydroxide (TMAOH). Then, a solution of Dox (2 mg mL^−1^) was mixed with Ti_3_C_2_ nanosheets to load the former onto their surface by adsorption. Additionally, to improve the targetability of the nanostructures, the surface of Ti_3_C_2_ MXene was further modified by Hyaluronic acid (HA) through adsorption, which also serves as a capping agent (zeta potential = −20.71 mV). In vitro studies were conducted on HCT-116 cells (a human colon carcinoma cell line) to assess the therapeutic efficacy of Ti_3_C_2_ nanosheets. The photothermal effect of Ti_3_C_2_ irradiated nanosheets (without Dox to avoid fluorescence interference with FITC) on HCT-116 cells was examined using flow cytometry. Photothermal groups were irradiated with a laser at 808 nm and 0.8 W/cm^2^ for 10 min. Figure 6 shows a significant difference between cells treated with laser or Ti_3_C_2_ separately and those treated with a combination of laser and Ti_3_C_2_. Based on Figure 6, it is believed that cell death was caused by late apoptosis and was dose-dependent, specifically with respect to the Ti_3_C_2_ nanosheets. The cytotoxic viability studies conducted by the MTT assay showed no significant difference in cell viability in the groups treated with Dox only and Dox-loaded Ti_3_C_2_ nanosheets, as shown in Figure 6b. However, cells treated with Ti_3_C_2_-Dox and laser irradiation (808 nm, 0.8 W/cm^2^) resulted in the highest cell killing due to the PTT and chemotherapy synergy. The authors also suggested that Ti_3_C_2_-Dox nanosheets can act as a photosensitizer, producing reactive oxygen species (ROS) upon irradiation with an 808 nm laser. This was validated by an ROS probe. The oxidation of the nonfluorescent 2′-7′dichlorofluorescin diacetate (DCFH-DA) produces 2′-7′dichlorofluorescein (DCF), which emits green fluorescence (λ = 530 nm) at an excitation wavelength of λ = 485 nm [37]. DCFH-DA can penetrate cell membranes where two acetate groups get hydrolyzed by intracellular esterases to produce DCFH, which reacts with ROS to produce fluorescent DCF [37,38].

Liu et al. [25] confirmed the formation of ROS upon irradiating HCT-116 cells that were incubated with Ti_3_C_2_-Dox and DCFH-DA (λ = 808 nm, 0.8 W/cm^2^, 10 min), where samples were excited at 488 nm and images acquired at 550 nm. The green fluorescence of DCF is shown in Figure 7. Fluorescence images also showed the specific targeting ability of the HA on Ti_3_C_2_-Dox nanosheets in CD44^+^-overexpressed tumor cells (HCT-116), as indicated by the fluorescence of Dox. As expected, the low fluorescence of Dox was detected in A2780 cells due to the absence of CD44^+^.

The in vivo studies confirmed the synergistic PTT/PDT/chemotherapy. As shown in Figure 8a,b thermal images revealed localized heating in the tumor, reaching ~53 °C within 5 min of NIR irradiation. The tumor was completely eradicated when treated with Ti_3_C_2_-Dox+laser, while other groups reflected an increase in tumor growth, as shown in Figure 8c,d [25].

Recently, MXenes garnered attention in nanotheranostics, especially in cancer treatment, due to the unique optical, electronic, and thermal properties that MXenes exhibit. An et al. [39] fabricated MRI-guided manganese-functionalized Ti_3_C_2_ nanosheets for use in cancer treatment. Those nanosheets were synthesized according to the HF etching method and were modified with PEG to improve their biocompatibility. In addition, the surface of Ti_3_C_2_ was further modified by attaching manganese (Mn) ions to endow magnetic and Fenton-like catalytic properties. The former feature serves as a contrast agent in T_1_-weighted MRI, and the latter in chemodynamic therapy (CDT). CDT leverages the acidic tumor microenvironment to generate ROSs through Fenton-like reactions to induce cell apoptosis [40]. Mn ions act as a catalyst in the Fenton reaction, where hydrogen peroxide (H_2_O_2_) reacts to form highly toxic hydroxyl free radicals (${HO}^{•}$) that induce apoptosis and kill cancer cells [41]. H_2_O_2_ is a product of mitochondrial oxidative respiration, overproduced in cancer cells [41,42]. The Mn-Ti_3_C_2_@PEG MXene synthesized by An and coworkers [39] exhibited PTT-CDT anticancer synergistic effect. In addition, Mn ions acted as an MRI contrast agent. In vivo studies on 4T1 tumor-bearing mice showed a complete tumor eradication for the group treated with combined Mn-Ti_3_C_2_@PEG (20 mg kg^−1^) and laser irradiation (λ = 808 nm, 1.5 W/cm^2^, 10 min) due to the synergistic PTT-CDT, as shown in Figure 9a. Tumor ablation by the PTT effect was confirmed by the infrared thermal images, which displayed an increase in temperature up to 48 °C for the sample treated with irradiated Mn-Ti_3_C_2_@PEG MXene, as shown in Figure 9b. The CDT effect was validated by the intracellular generation of hydroxyl radicals, as detected by DCFH-DA in vitro studies on 4T1 cells incubated with Mn-Ti_3_C_2_@PEG in the presence or absence of H_2_O_2_. Using DCFH-DA, a green fluorescence was detected under a fluorescence microscope, thus indicating the generation of ROS (${HO}^{•}$). In addition, Mn ions improved the quality of MRI images as detected by the increased brightness level after 10 min of IV injection of Mn-Ti_3_C_2_@PEG.

Zong et al. [43] fabricated GdW_10_@Ti_3_C_2_ composite nanosheets integrating GdW_10_-based polyoxometalates (POMs) into Ti_3_C_2_ nanosheets for multifunctional therapeutic and diagnostic use. Gadolinium (Gd) is a contrast agent for T_1_-weighted magnetic resonance imaging (MRI), whereas tungsten (W) can serve as a contrast agent in computed tomography. The combination of the photothermal therapy of Ti_3_C_2_ nanosheets with the contrast agents for CT/MRI provides a potential tool for image-guided tumor ablation. The T_1_-weighted MRI images of 4T1 tumor-bearing mice were enhanced using GdW_10_@Ti_3_C_2_ and reflected a gradual increase in the accumulation of the nanosheets at the tumor site in mice due to the EPR effect at 0–2 h post-injection. The CT images also showed enhanced brightness at the tumor site after intertumoral (i.t.) and intravenous (i.v.) injections. The therapeutic efficacy of GdW_10_@Ti_3_C_2_ was also investigated in vivo. Thermal images of 4T1 tumor-bearing mice treated with GdW_10_@Ti_3_C_2_ showed a high increase in temperature up to 56 °C after 10 min of laser irradiation at 808 nm and 1.5 W/cm^2^. In addition, the tumor growth was completely inhibited by hyperthermia (with no recurrence) in the group treated with laser and GdW_10_@Ti_3_C_2_ compared to the groups receiving other treatments (control, laser only, GdW_10_@Ti_3_C_2_ only).

MXene in the form of quantum dot (MQD) was found to act as a photosensitizer (PS) and a photothermic agent (PTA) according to Feng et al. [44]. Therefore, it can be used in photothermal and photodynamic therapies. Feng and coworkers fabricated a multifunctional nanoparticle (MQD@ZIF-8) composed of zeolitic imidazolate framework-8 (ZIF-8) and MXene quantum dot (MQD). ZIF-8 is a low-toxicity metallic organic framework (MOF) that can efficiently carry drugs and other nanoparticles, such as MQD, especially as the latter is unstable in solutions due to aggregation. In this study, Feng et al. [44] loaded doxorubicin in MQD@ZIF-8 nanoparticles to investigate the anti-tumor effect of combined chemotherapy and phototherapy in HeLa cells. A positive correlation was found between the concentration of MQD@ZIF-8 and the temperature increase when an aqueous solution is irradiated by the NIR laser. For example, the temperature increased from ~42 °C to 53 °C when the concentration of MQD@ZIF-8 was increased from 0.25 to 0.5 mg mL^−1^ under 808 nm NIR laser irradiation at 2 W/cm^2^ for 5 min [44]. Power density of the NIR laser source was also found to be directly correlated to temperature increase. These findings show that the MQD@ZIF-8 nanoparticle has the potential to kill cancer cells via PTT. In addition, the generation of singlet oxygen ^1^O_2_ from MQD@ZIF-8 nanoparticle was verified by UV light irradiation in the presence of DCFH-DA. The green fluorescence captured in confocal fluorescence images was due to the oxidation of DCFH-DA by ^1^O_2_ to fluorescent DCF. Hence, the fabricated nanoparticle has the potential to serve in PDT. Feng et al. [44] also conducted in vitro experiments using HeLa cells to examine the cytotoxicity of two formulations: MQD@ZIF-8 and MQD@ZIF-8/Dox in response to NIR (808 nm laser) and UV light. Both formulations showed a significant decrease in cell viability when treated with NIR compared to control groups (without irradiation). Moreover, irradiating the MQD@ZIF-8 formulation with both NIR and UV has further decreased the cell viability compared to the non-irradiated group and those receiving a single dose of irradiation (NIR or UV). For further studies implementing MXenes in cancer treatment, refer to Table 1. These studies mainly use photothermal (PTT) and/or sonodynamic (SDT) therapy as a triggering mechanism for MXenes.

## 4. Biocompatibility, Toxicity, and Stability of MXenes

Biocompatibility, a major aspect of any drug delivery system, mandates that the administered biomaterials do not exhibit harm nor cause adverse effects to the patient’s health while serving their intended medical function. Biocompatibility and toxicity are often used interchangeably despite their different implications. Toxicity measures unwanted damage to cells, organs, or the patient [56], and the term is often used by the pharmaceutical industry. Biocompatibility is a term used in biomedical applications and was defined in 1986 as “the ability of a material to perform with an appropriate host response in a specific application” [57]. Therefore, biocompatibility encompasses both the potential toxicity of the biomaterial in the body and the effect of the physiological environment on its performance [56]. The biocompatibility of MXenes is influenced by their size, structure, surface modification, dose, exposure time, and the method of synthesis [58,59].

Several studies have demonstrated the good compatibility of MXenes in vitro when administered at a proper dosage; however, in vivo studies are scarce in the literature. To improve the biosafety of MXenes, they can be functionalized with biocompatible materials (e.g., PEG, PVP, HA, soybean, etc.) [25,47,48,49]. Dai et al. [60] studied the biocompatibility of soybean modified nanosheets of MXene (MnO_x_/Ta_4_C_3_-SP) administered at three doses of 5, 10, and 20 mg kg^−1^. According to various blood indexes (white and red blood cells, platelets, hemoglobin, lymphocytes, etc.) evaluated 30 days post-injection, all the groups that received the treatment showed no abnormalities compared to those of the non-particle control group. In addition, hematoxylin and eosin (H&E) staining of key organs (heart, liver, spleen, kidney, lung) reflected no histological abnormalities, and thus indicated a good biosafety of the tested dosages. Xing and coworkers [45] encapsulated MXenes along with DOX in cellulose hydrogel for dual chemo-phototherapy. Without irradiation, the intratumoral injection of different hydrogel treatments in mice (hydrogel alone, DOX and/or MXene-loaded hydrogels) resulted in no adverse effects in organs such as the heart, liver, spleen, lung, and kidney according to a histological analysis 15 days post-injection. In addition, protein levels of proinflammatory cytokines in serum remained unchanged in all the groups treated with hydrogels as opposed to the one treated with free DOX. Cells exposed to MXenes can be damaged by mechanical and/or oxidative stress. Rozmysłowska-Wojciechowska et al. [61] found that coating Ti_3_C_2_ and Ti_2_C MXenes with collagen increased their biocompatibility compared to pristine MXenes according to cell viability analysis for MXene concentrations of 1–125 mg L^−1^. Coating MXenes with collagen reduced oxidative stress, as evidenced by the decrease in ROS production after 24 h of incubation, with a more pronounced effect observed in the MCF-7 cell line exposed to collagen-coated Ti_2_C. This also implies that MXenes’ performance depends on the type of cell line under examination.

Controlling the size of MXenes is another approach to improve their biocompatibility [62,63]. Shao et al. [64] fabricated nitride-based MXene quantum dots (Ti_2_N QDs) with a size of 5 nm. According to the in vivo biodistribution study at a dose of 20 mg kg^−1^, high concentrations of Ti_2_N QDs were found 4 h post-injection in the tumor, kidney, and liver compared to other organs. It was anticipated that the EPR effect was behind the high uptake of Ti_2_N QDs in the tumor. On the other hand, the high accumulation in the liver and kidney could be attributed to the RES clearance and renal excretion. This was validated by the amount of Ti_2_N QDs that were cleared from the body 6–18 h post-injection (integral masses of 15.6% and 12.2% in feces and urine). In vivo biocompatibility of Ti_2_N QDs (20 mg kg^−1^) was further assessed 20 days post-injection, and both the blood biochemistry assay and H&E of major organs showed no abnormalities compared to the control group. These findings support the successful biodegradation (to TiO_2_) of this type of MXene in the body.

Although the small size of Ti_2_N MXenes (<5 nm) increases their renal excretion from the body, as reported by Shao et al. [64], other families of small MXenes can show higher toxicity according to Dmytriv and Lushchak [58]. Small-sized and delaminated MXenes can cause non-specific damage because of their sharp edges, resembling the effect of “nanoknives” on the surface of cells [58]. This causes cytoplasm to leak from cells and ultimately collapse. Therefore, large and non-delaminated MXenes reduce the physical effect of “nanoknives” and hence improve their biocompatibility. And while small-size MXenes (<5 nm) are removed mainly by renal excretion, nanomaterials above 200 nm are excreted from the body through feces using lysosomal degradation by Kupffer cells and probably with the aid of myeloperoxidase enzyme to oxidize them [58,65]. This is further facilitated by the hydrophilic nature of MXenes, which enables them to destabilize and degrade upon interaction with an aqueous medium [58].

One cannot report the biocompatibility or toxicity of a certain medicine without referring to its safe dosage limit [63,66]. Han et al. [28] examined the biocompatibility of soybean MXene nanosheets (Ti_3_C_2_-SP) administered intravenously into mice at the following doses: 6.25, 12.5, 25, 50 mg kg^−1^. After 7 days post-injection, histocompatibility assay using H&E staining of major organs (heart, liver, spleen, lung, and kidney) showed no pathological changes as compared with the control group. To ensure the safe excretion of the nanosheets from the body, the amount of Ti content in urine and feces was monitored using an ICP-OES test. At a dosage of 15 mg kg^−1^ of Ti_3_C_2_-SP, urine and feces were collected after 2, 6, 12, 24, and 48 h. Authors found that the total amount of Ti excreted in urine and feces reached 18.70% and 10.35%, respectively, thus suggesting their safe elimination from the body. In another study, Jastrzebska et al. [66] assessed the cytotoxicity of Ti_3_C_2_ MXene sheets on two normal (MRC-5 and HaCaT) and two cancerous (A549 and A375) cell lines using MTT and calcein-AM assays after 24 h of incubation. Different concentrations of Ti_3_C_2_ solutions were tested up to 500 mg L^−1^. Cell viability using the MTT assay showed that the cytotoxicity of Ti_3_C_2_ increased with dosage for both normal and cancerous cell lines. The viability of HaCaT cells remained above 80% for all the tested concentrations, while A549 experienced the highest change in viability (<25% at 500 mg L^−1^). Thus, the authors concluded that the cytotoxicity of Ti_3_C_2_ is dependent on the dosage and the cell line, with a higher toxicity against cancerous cells as compared to normal ones.

The stability of MXenes is crucial for therapeutic applications. MXenes tend to degrade at room temperature when exposed to light, water, and oxygen [67]. For example, the most commonly used MXene Ti_3_C_2_T_x_, oxidizes rapidly when stored in an aqueous solution open to air. Zhang et al. [68] found that Ti_3_C_2_T_x_ MXene solutions degraded by 42%, 85%, and 100% when stored in open vials for 5, 10, and 15 days, respectively. The degraded solution was characterized by a cloudy-white colloidal suspension containing primarily anatase (TiO_2_). Branches of anatase were found to grow on the edges of the nanosheets upon degradation as detected in the TEM images. In addition, the degradation of the nanosheets was size-dependent; the smaller they were, the more prone they were to oxidation. To improve the stability of Ti_3_C_2_T_x_ in an aqueous solution, the group displaced dissolved oxygen with argon at low temperature. It was found that eliminating the main oxidant (O_2_) had a more pronounced effect on the stability of MXene than lowering the temperature, with the best stability achieved by combining both strategies. Zhang et al. [68] fitted the degradation results of Ti_3_C_2_T_x_ solution with the exponential decay model and estimated that the shelf life can be extended up to 2 years when stored in hermetic Ar-filled bottles at 5 °C. Consequently, the authors recommended that the ideal conditions for extending the lifetime of MXene colloidal solutions are large nanosheets stored under argon at low temperatures. Filtrating MXene flakes from water is another approach to improve their stability upon storage.

Water is a potential source of oxidation for MXenes. Maleski et al. [69] examined the stability of Ti_3_C_2_T_x_ MXene dispersions in several organic solvents. The long-term stability after 40 days of storage away from sunlight was maintained in DMF, NMP, PC, and ethanol solutions compared to water. The authors observed the formation of white sediments of TiO_2_ in the Ti_3_C_2_T_x_ MXene when stored in water, thus confirming its oxidation. Although the degradation could be ascribed to the reaction of MXene with dissolved oxygen in water, oxygen is not the sole factor, knowing that its solubility is at least an order of magnitude higher in DMF, NMP, PC, and ethanol [69]. Therefore, the authors concluded that Ti_3_C_2_T_x_ MXene degradation is due to a combination of water hydrolysis and a mix of water and oxygen, rather than dissolved oxygen alone. Similar findings were reported by Huang et al. [70]. It was found that MXenes would oxidize faster in the presence of both O_2_ and H_2_O (time constant of ~5 days for Ti_3_C_2_-H_2_O/O_2_). However, H_2_O had a greater effect on MXene’s stability compared to O_2_, as the time constant was ~41 and 2026 days for Ti_3_C_2_-H_2_O/Ar and Ti_3_C_2_-iso-propanol/O_2_, respectively. The stability was further improved to ~4753 days in the absence of both H_2_O and O_2_ (Ti_3_C_2_-iso-propanol/Ar). Besides water and oxygen, MXene’s degradation is also affected by temperature, pH of the medium, and the surface terminators of its own structure. For more details, refer to [58,71]. The degradation of MXenes in biological media can be improved by surface modification using stabilizing agents including antioxidants (such as ascorbic acid and sodium ascorbate), synthetic polymers (such as PVP and PVA), and natural polymers (such as soybean and cellulose) [58]. Embedded ROS scavengers, such as silica/gold nanoparticles, and pH-responsive coatings, such as polydopamine, can improve the long-term stability of MXenes [71]. Jastrzębska and coworkers [72] assessed the stability of 2D V_2_CT_z_ MXene in cell culture medium (DMEM). Authors reported a lower zeta potential value for V_2_CT_z_ MXene in DMEM compared to double-distilled water (−6.0 vs. −18.8 mV). The appearance of the DMEM corona proteins on the surface of the MXene flakes has probably caused the lower zeta potential value in DMEM. After 24 h of incubation in DMEM, the zeta potential value dropped to −2.8 mV due to the oxidation of MXene to precipitating V_x_O_y_ oxides. Similar findings were reported by Wu et al. [73] for Ti_3_C_2_T_x_ in human plasma.

MXenes are gaining attention in biomedical applications. However, most existing studies have focused on evaluating their biocompatibility or toxicity based on a single factor. To ensure the biosafety of MXenes, further comprehensive research is needed that considers multiple variables such as concentration, size, structure, dosage, and exposure duration across different cell lines. Although most of the studies mentioned earlier suggested good biocompatibility, some conflicting findings in the literature reported potential side effects such as respiratory disorders [74], inhibition of embryo angiogenesis, and deregulation of genes responsible for cell proliferation, survival, cell death, and angiogenesis [75]. Additionally, current research primarily addresses short-term effects, so it is essential to explore the long-term impacts as well. Since MXenes may accumulate in certain organs, in-depth biodistribution studies are necessary to evaluate their long-term toxicity and biodegradability. Therefore, a thorough understanding of MXenes’ safety profile will support their broader application in the biomedical field, especially due to their distinctive and tunable properties.

## 5. Conclusions

In this review, MXenes were introduced, and their potential use in cancer treatment was detailed. MXenes are carbides and/or nitrides of transition metals that belong to 2D materials, with titanium carbide (Ti_3_C_2_) being the most used type of MXenes in research. MXenes are often fabricated by etching a MAX phase with HF acid, thus resulting in a negatively charged surface that can be used to load cationic drugs such as doxorubicin with a high loading capacity. MXenes can also convert light to heat when irradiated in the NIR region, exhibiting high photothermal conversion efficiency, thereby making them excellent photothermal agents in photothermal therapy, which kills cancer cells through hyperthermia. The synergy between PTT and chemotherapy is often used in drug delivery systems. Moreover, the flexibility of modifying the surface of MXenes enables their use in photodynamic and chemodynamic therapy to generate reactive oxygen species that eventually help in fighting cancer cells. Contrast agents for MRI and CT can also be incorporated in MXenes, which serve in image-guided therapies.

Despite the advantages of using MXenes in cancer treatment, there are a few concerns related to their fabrication by HF etching. HF acid is highly toxic to humans by inhalation, ingestion, and skin absorption, even at low concentrations. This issue can be resolved by the in situ HF etching method, but it usually requires a much longer duration to fully etch the MAX phase. The biosafety of MXenes to normal cells is another issue that needs careful investigation with respect to various factors. The potential use of MXenes in targeted drug delivery systems is another field to explore in the future, with the possibility of loading them into other DDSs as a photothermal agent (e.g., MXenes-encapsulated liposomes for enhanced PTT).

## Figures and Tables

**Figure 1 ijms-26-10296-f001:**
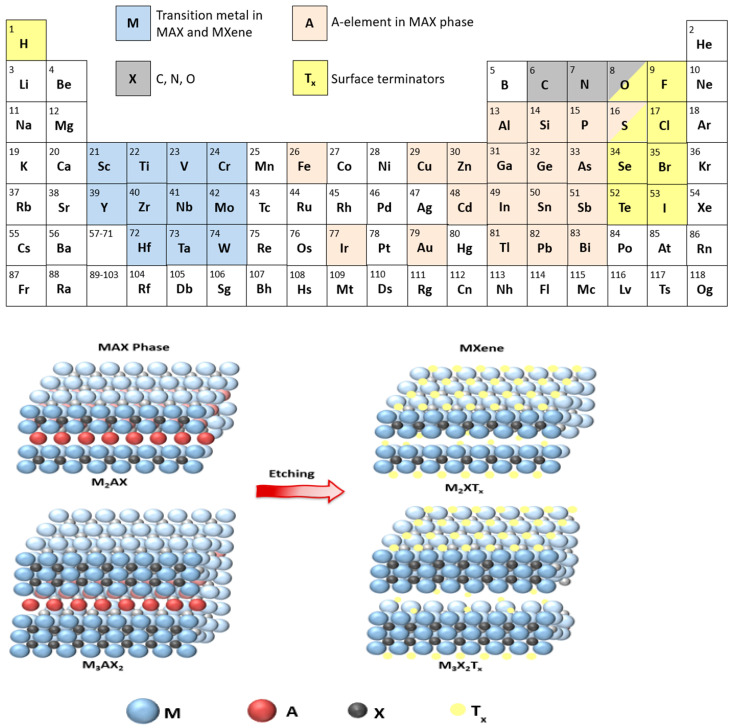
Chemical etching of the MAX phase to produce MXene with elemental constituents of both highlighted in the periodic table.

**Figure 3 ijms-26-10296-f003:**
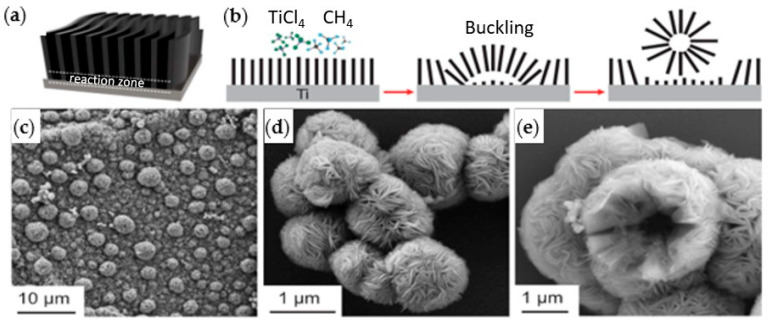
(**a**) Schematic diagram of the carpet-like structure growth of Ti_2_CCl_2_ MXene, (**b**) reaction zone and vesicle formation and detachment. SEM images for vesicle formation (**c**) and detached micro-vesicles (**d**,**e**); reproduced from [24] with permission from the American Association for the Advancement of Science.

**Figure 4 ijms-26-10296-f004:**
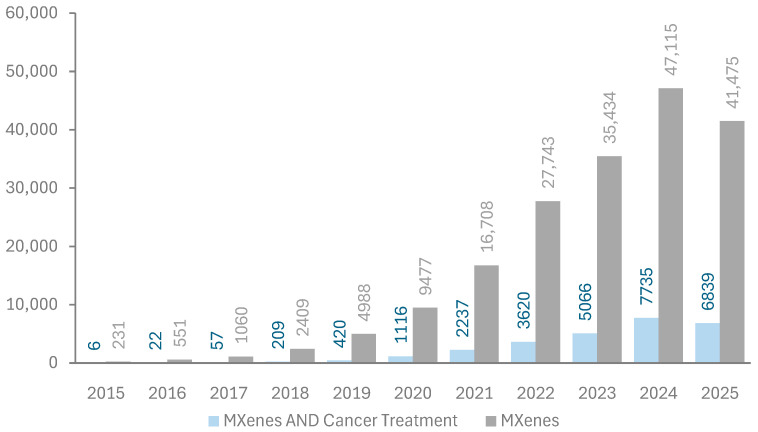
Publication metrics per year on MXenes. Data extracted from the Dimensions database [30]. Keywords used in the search are “MXenes” and “MXenes AND Cancer treatment”.

**Figure 5 ijms-26-10296-f005:**
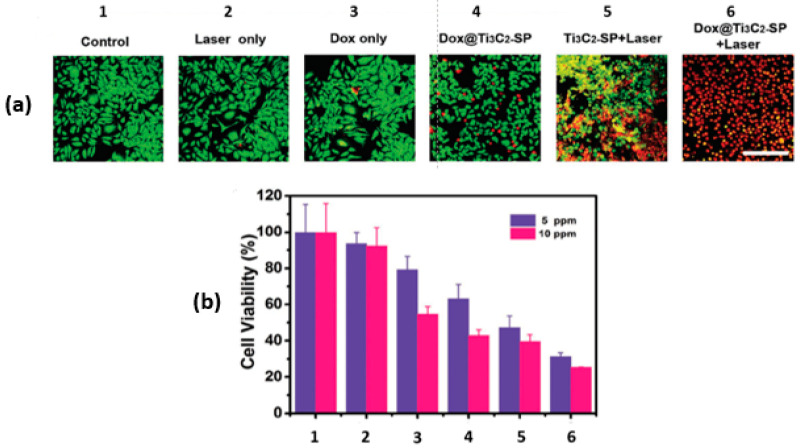
In vitro intracellular drug delivery. (**a**) CLSM images of 4T1 cells; (**b**) 4T1 cell viabilities after different treatments (1 to 6) at two concentrations; reproduced from [28] with permission from WILEY-VCH Verlag GmbH & Co.

**Figure 6 ijms-26-10296-f006:**
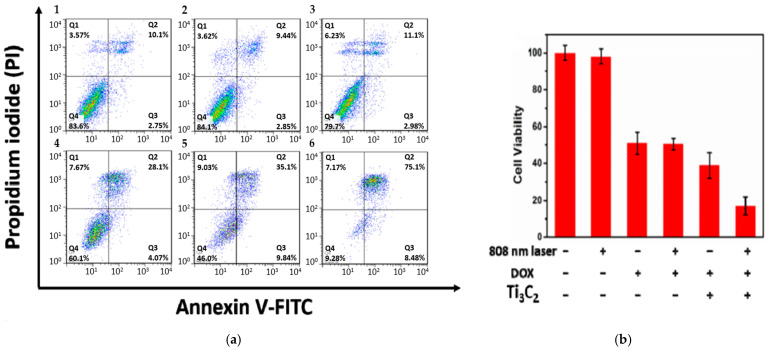
(**a**) flow cytometry of HCT-116 cells treated with (1) control, (2) laser only (λ = 808 nm, 0.8 W/cm^2^, 10 min), (3) Ti_3_C_2_ nanosheets (75 µg mL^−1^) in dark, and combined laser irradiation with different concentrations of Ti_3_C_2_ nanosheets: 25, 50, and 75 µg mL^−1^ (4–6). (**b**) MTT assay of HCT-116 cells with Ti_3_C_2_ and Dox concentrations of 50 and 42 µg mL^−1^, respectively. Reproduced from [25] with permission from the American Chemical Society.

**Figure 7 ijms-26-10296-f007:**
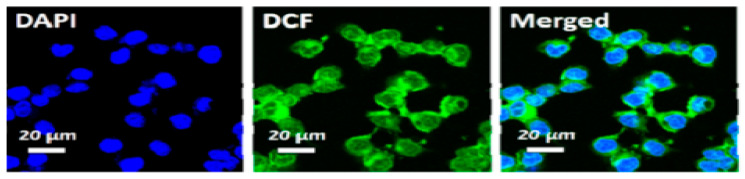
Intracellular ROS generation upon irradiation with NIR (λ = 808 nm, 0.8 W/cm^2^, 10 min) in HCT-116 cells incubated with Ti_3_C_2_-Dox; reproduced from [25] with permission from the American Chemical Society.

**Figure 8 ijms-26-10296-f008:**
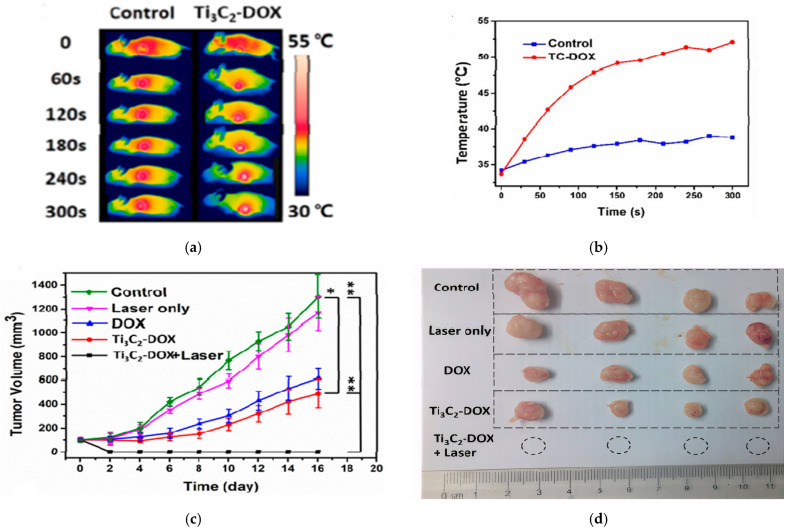
(**a**) thermal images of mice (control and Ti_3_C_2_-Dox) under different irradiation times, (**b**) temperature profile of the tumor, (**c**) tumor growth for different treatments (n = 4, mean ± SD, * *p* < 0.05, ** *p* < 0.01), (**d**) photos of tumor size after treatment; reproduced from [25] with permission from the American Chemical Society.

**Figure 9 ijms-26-10296-f009:**
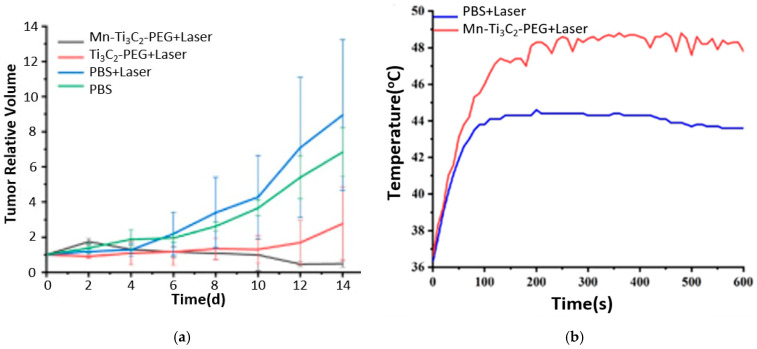
(**a**) Tumor growth curves, (**b**) temperature profile at tumor from infrared thermal pictures for in vivo studies; reproduced from [39] with permission from PubMed Central.

**Table 1 ijms-26-10296-t001:** MXene studies under investigation for cancer treatment.

MXene Type	Treatment Modality	Drug	Cancer Cell Line	Major Outcomes	Ref
Ti_3_C_2_	▪ Ti_3_C_2_-HA nanosheets ▪ Synergistic chemotherapy/photodynamic/photothermal (NIR-I: 808 nm) therapy	Doxorubicin	HCT-116 human colon carcinoma cancer cells (in vitro and in vivo)	▪ Significant reduction in HCT-116 cell viability in vitro for combined chemo-photothermal treatment (Ti_3_C_2_-Dox+Laser) compared to each treatment separately ▪ Complete tumor eradication in vivo for Ti_3_C_2_-Dox+laser group compared to other treatment groups, showing sustained tumor growth ▪ Ti_3_C_2_ produces ROS under NIR irradiation, a potential use in PDT	[25]
Ti_3_C_2_	▪ Ti_3_C_2_-SP nanosheets ▪ Synergistic chemotherapy/photothermal (NIR-I: 808 nm) therapy	Doxorubicin	4T1 breast cancer cells (in vitro and in vivo)	▪ Reduced 4T1 cell viability in vitro for combined chemo-photothermal treatment (Dox@Ti_3_C_2_-SP+Laser) compared to each treatment separately ▪ Complete tumor eradication in vivo for Dox@Ti_3_C_2_-SP+laser group compared to 80.3% tumor inhibition (with recurrence) for Ti_3_C_2_-SP+Laser	[28]
Ti_3_C_2_	▪ Ti_3_C_2_ cellulose hydrogels ▪ Synergistic chemotherapy/photothermal (NIR-I: 808 nm) therapy	Doxorubicin	SMMC-7721, HepG2, U-118MG, U-251MG (in vitro) SMMC-7721 and HepG2 (in vivo)	▪ Reduced toxicity and improved biocompatibility due to hydrogel shielding of MXene ▪ Reduced cell viability (<20% for HepG2) in vitro for photothermal treatment of cellulose/MXene hydrogels (1 W/cm^2^, 5 min) ▪ PTT (MXene hydrogels) is more effective than chemotherapy (Dox hydrogels) in vivo, but with tumor relapses for the former ▪ Complete tumor eradication in vivo for combined Dox-MXene hydrogels when irradiated at 1 W/cm^2^ for 5 min	[45]
Ti_3_C_2_	▪ Spikey and biodegradable spikey MXene nanosheets (sMXene and dsMXene) ▪ Synergistic chemotherapy/photothermal (NIR-II: 1064 nm) therapy	Doxorubicin	KHOS cells (in vitro and in vivo)	▪ Biocompatible MXenes, sMXenes and dsMXenes (without Dox) ▪ Significant reduction in KHOS cell viability in vitro for Dox-loaded dsMXene compared to Dox+MXene/sMXene ▪ Excellent anticancer ability in vitro for Dox-loaded dsMXene when irradiated at 2 W/cm^2^ for 10 min ▪ Tumor inhibition rate in vivo was the highest for the group treated with Dox-loaded dsMXene+NIR	[46]
Ti_3_C_2_	▪ Targeted therapy: MUC1 Aptamer (Apt-M) conjugated to Ti_3_C_2_ ▪ Photothermal therapy (NIR-I: 808 nm)		MCF-7 breast cancer cells (in vitro and in vivo)	▪ Reduced MCF-7 cell viability in vitro for PTT compared to nonirradiated treatments: 43.1% for Ti_3_C_2_-PEG, and 22.7% for Ti_3_C_2_/Apt-M ▪ Reduced MCF-7 cell viability in vitro due to targeting Ti_3_C_2_/Apt-M+laser compared to controls (Laser+Ti_3_C_2_-PEG or Ti_3_C_2_/Apt-C) ▪ Suppressed tumor size effectively in vivo with a tumor weight significantly lower in Ti_3_C_2_/Apt-M+laser group compared to other treatment groups over 17 days ▪ Biocompatible Ti_3_C_2_/Apt-M at a dose of 15 mg kg^−1^ (in vivo)	[47]
Ti_3_C_2_	▪ TiO_2-X_ as a sonosensitizer in Ti_3_C_2_@TiO_2-X_-PEG (TTP) ▪ Synergistic sonodynamic/photothermal (NIR-II: 1064 nm) therapy		4T1 breast cancer cells (in vitro and in vivo)	▪ Reduced viability for 4T1 cells treated with TTP in vitro: 45% and 39% after individual PTT and SDT, respectively, and 16.5% after combined PTT/SDT ▪ Complete tumor eradication after 14 days in the combined laser & US (PTT & SDT) group compared to other treatments ▪ Biocompatible TTP at ~40 mg kg^−1^ (in vivo)	[48]
Ti_3_C_2_	▪ Ti_3_C_2_-PVP nanosheets ▪ Synergistic iron chelation/chemotherapy/photothermal (NIR-I: 808 nm) therapy	DOXjade: deferasirox tethered to doxorubicin via a hydrazone bond	HCT116 colorectal cancer cells (in vitro and in vivo)	▪ 5.2- and 1.7-fold improvement in IC_50_ value for Ti_3_C_2_-PVP@DOXjade irradiated at 808 nm (1 W/cm^2^) relative to DOXjade and Ti_3_C_2_-PVP+laser treatments, respectively ▪ Significant tumor eradication, 2 weeks post treatment, with no recurrence for groups treated with Ti_3_C_2_-PVP@DOXjade+laser ▪ Biocompatible Ti_3_C_2_-PVP at a dose of 15 mg kg^−1^ (in vivo)	[49]
Ti_3_C_2_	▪ Heat-treated H-Ti_3_C_2_-PEG nanosheets ▪ Photothermal (NIR-II: 1064 nm)-enhanced sonodynamic therapy		4T1 breast cancer cells (in vitro and in vivo)	▪ Heat treatment of Ti_3_C_2_ formed sonosensitizer TiO_x_ on the surface of nanosheets, which improved the ROS generation efficiency by facilitating the electron-hole separation ▪ Reduced viability for 4T1 cells treated with H-Ti_3_C_2_-PEG in vitro up to ~13.4% after combined PTT/SDT compared to ~79.5% and 53.4% after individual PTT and SDT, respectively ▪ Complete tumor eradication with no recurrence for groups treated with H_H_-Ti_3_C_2_-PEG+laser+US compared to ~54.9% tumor inhibitory rate for H_H_-Ti_3_C_2_-PEG + US. Negligible tumor suppression for other treatment groups ▪ Biocompatible H_H_-Ti_3_C_2_-PEG at a dose of 20 mg kg^−1^ (in vivo)	[50]
Ti_3_C_2_	▪ Sonosensitizer and peptide onto Ti_3_C_2_ nanosheets (Ar-Ti_3_C_2_-TiO_2_-RGD) ▪ Photothermal (NIR-I: 808 nm)-enhanced sonodynamic therapy		4T1 (in vitro and in vivo) and MCF-7 (in vitro) breast cancer cells	▪ Increased US-triggered ROS generation in Ar-Ti_3_C_2_-TiO_2_ compared to pure TiO_2_ due to reduced band gap and improved electron-hole separation ▪ Reduced viability for 4T1 cells treated with Ar-Ti_3_C_2_-TiO_2_-RGD in vitro (~10.1%) after combined PTT/SDT compared to ~23.3% and 33.6% for individual PTT/SDT, respectively ▪ Tumor inhibition rate in vivo was the highest (92.92%) for groups treated with Ar-Ti_3_C_2_-TiO_2_-RGD+laser + US compared to other treatments	[51]
Ti_3_C_2_T_x_	▪ MXene and/or doxorubicin-loaded supramolecular hydrogel (MGel, DGel, MDGel) ▪ Synergistic chemotherapy/photothermal (NIR-I: 808 nm) therapy	Doxorubicin	B16F10 murine melanoma cells (in vitro and in vivo)	▪ Significant reduction in B16F10 cell viability in vitro for dual-function hydrogel systems with NIR irradiation (MGel/DGel/MDGel+laser) compared to non-irradiated treatments ▪ Synergistic chemotherapy/PTT. Significant cytotoxicity in vitro for the groups treated with irradiated MDGel (77.2 ± 2.9%) compared to irradiated MGel/DGel (45.7 ± 0.6% and 30.1 ± 0.3%, respectively) ▪ Tumor inhibition rate in vivo was the highest (68.9 ± 6.9%) for synergistic chemotherapy/PTT (MDGel+laser) compared to other treatments	[52]
Ti_3_C_2_T_x_	▪ Photothermal therapy (NIR-I: 808 nm)		Conventional and 3D-bioprinted spheroids using 4T1 breast cancer cells	▪ Reduced cell viability after MXene administration & PTT (44.7% for bioprinted spheroids treated with multi-layer MXene (MX) and 43.1% for conventional spheroids treated with few-layer MXene (FX) ▪ Increased ROS production due to NIR irradiation	[53]
Ti_2_C	▪ PEG-coated Ti_2_C nanosheets ▪ Photothermal therapy (NIR-I: 808 nm)		Malignant MCF-7 breast cancer cells, malignant A375 melanoma, non-malignant MCF-10A, and normal HaCaT skin cells (in vitro)	▪ Reduced viability in malignant cells compared to non-malignant cells at a dose of 37.5 μg mL^−1^ of Ti_2_C-PEG after PTT treatment ▪ Effective reduction in therapeutic dose compared to other studies (24 times lower) ▪ Biocompatible Ti_2_C-PEG (in vitro)	[54]
Nb_2_C	▪ PVP-coated Nb_2_C nanosheets ▪ Photothermal therapy (NIR-I&II: 808 and 1064 nm)		4T1 breast cancer cells (in vitro and in vivo) and Glioma U87 cancer cells (in vitro)	▪ Reduced viability (<20%) for 4T1 and U87 cells treated with Nb_2_C at a power density of 1 W/cm^2^ (NIR-I and II) compared to control groups ▪ Complete tumor eradication, 2 weeks post treatment, with no recurrence for groups treated with Nb_2_C-PVP+laser (NIR-I and II) ▪ Biocompatible Nb_2_C-PVP at a dose of 20 mg kg^−1^ (in vivo)	[55]

## Data Availability

No new data were created or analyzed in this study. Data sharing is not applicable to this article.

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
