# Peer review of "The Emerging Role of MXenes in Cancer Treatment"

_ijms, 2025, doi:10.3390/ijms262110296_

Round 1
Reviewer 1 Report
Comments and Suggestions for Authors
MXenes are a class of 2D transition-metal carbides/nitrides (Mn+1XnTx), mostly synthesized through chemical etching of ceramic precursors. Their tunable catalytic, optical, and electronic properties have driven rapid growth in applications, particularly in energy storage and biosensing. Recent studies highlight MXenes’ potential in cancer therapy as they efficiently convert near-infrared light into heat for photothermal treatment and can be surface-modified with drugs or nanoparticles to enable synergistic photo-, chemo-, or sonodynamic therapies. Through this review, Salkho et al discuss several synthesis strategies, therapeutic applications, and current knowledge on MXene biocompatibility and cytotoxicity. The review has been articulately presented, however, there are some comments and suggestions that need to be addressed before being accepted by the journal IJMS.
- The Introduction section starts with the data by WHO. I believe the authors can find much recent information that 2022 to provide updated statistics.
- In page 2, where Mustard gas has been mentioned as the first chemotherapeutic agent, a reference needs to be provided for that info.
- The part of periodic table in page 2 has a typo with the element with atomic number 20. “Cs” needs to be replaced with “Ca”.
- In page 4, too much information has been added. It is trivial to provide data regarding the mass and volume used in the cited paper. It can simply be avoided by citing the article where the data is derived from without giving a detailed synthesis process.
- Figure 2 has doubly labeled alphabets as insets.
- Section 2.3 could be divided into subsections for clarity.
- Figure 5(b) is difficult to understand without the x-axis labels. Nothing has been mentioned regarding it in the main text as well.
- Figure 6(a) is not clear at all. The copied data/figure is not visually understandable. Please put on a high-resolution image.
- I would expect to see more perspectives to the current developmental knowledge and strategies being approached by the researchers. These may include: immunological interactions, in vivo stability, scalability and reproducibility, biodegradability and clearance, toxicology and long term stability, and any other nanomechanical and translational challenges.
Reviewer 2 Report
Comments and Suggestions for Authors
In this manuscript, the authors gave a thorough review of progress and development of MXenes in the anti-tumor field. The authors demonstrated a comprehensive understanding of the field and cited literatures and gave a well-structured compilation of structures in the field. The manuscript is recommended for publication with the following minor revisions:
- The authors seem to dive in too deep with one example towards the end of the review. It is recommended to be more comprehensive than extremely focused.
- The disucssion on potential side effects and mechanisms is insufficient, which could be an interests to many readers in biology field.
Reviewer 3 Report
Comments and Suggestions for Authors
The manuscript submitted for review is an interesting review paper on one of the modern methods of destroying tumors. The review of available data on this issue was carefully prepared.
- This manuscript lacks an original approach to the topic, which could be demonstrated in the conclusions.
- My suggestions for this manuscript also apply to the graphic design. It is important that the authors pay attention to details such as the caption after the figure (see Figure 8). This caption should appear on the same page as the figure. Similarly, the title of Chapter 4 (line 466) should be moved to the next page. Table 1 in this manuscript is completely illegible. It would be advisable for the authors to improve the positioning of this table. Please consider changing the table to a horizontal orientation.
- I also suggest that the References section be prepared with greater care, particularly regarding the correct notation of chemical formulas.
Round 2
Reviewer 1 Report
Comments and Suggestions for Authors
I am satisfied with author's revision. Thank you.